materials science/electron microscopy/
electron microscopy

transmission electron microscopy,
*in situ* transmission electron microscopy,
liquid cell transmission electron microscopy

**Author for correspondence:**
Alex W. Robertson
e-mail: alex.w.robertson@gmail.com

This article has been edited by the Royal Society of Chemistry, including the commissioning, peer review process and editorial aspects up to the point of acceptance.

[†]These authors contributed equally to this work.

# Liquid cell transmission electron microscopy and its applications

Shengda Pu[†], Chen Gong[†] and Alex W. Robertson

Department of Materials, University of Oxford, Parks Road, Oxford OX1 3PH, UK

(iD) AWR, 0000-0002-9521-6482

Transmission electron microscopy (TEM) has long been an essential tool for understanding the structure of materials. Over the past couple of decades, this venerable technique has undergone a number of revolutions, such as the development of aberration correction for atomic level imaging, the realization of cryogenic TEM for imaging biological specimens, and new instrumentation permitting the observation of dynamic systems *in situ*. Research in the latter has rapidly accelerated in recent years, based on a silicon-chip architecture that permits a versatile array of experiments to be performed under the high vacuum of the TEM. Of particular interest is using these silicon chips to enclose fluids safely inside the TEM, allowing us to observe liquid dynamics at the nanoscale. *In situ* imaging of liquid phase reactions under TEM can greatly enhance our understanding of fundamental processes in fields from electrochemistry to cell biology. Here, we review how *in situ* TEM experiments of liquids can be performed, with a particular focus on microchip-encapsulated liquid cell TEM. We will cover the basics of the technique, and its strengths and weaknesses with respect to related *in situ* TEM methods for characterizing liquid systems. We will show how this technique has provided unique insights into nanomaterial synthesis and manipulation, battery science and biological cells. A discussion on the main challenges of the technique, and potential means to mitigate and overcome them, will also be presented.

## 1. Introduction

Visualizing liquid phase processes at the nanoscale can yield essential information for understanding fundamental processes in physics, chemistry and biology. Thanks to the development of liquid cell transmission electron microscopy (TEM) within the last 30 years, phenomena such as nanomaterial synthesis [1–6], live biological cells [7–12], battery solid–electrolyte interface (SEI) formation [13–15] and localized corrosion [16,17] have been visualized with unprecedented spatial and temporal resolution.

Liquid cell TEM allows the observation of processes that cannot be imaged with conventional TEM or other techniques. For example, biological specimens are destroyed within seconds in conventional TEM due to the high vacuum. And while cryo-electron microscopy can be used to safely image and determine their structures [18–20], the specimens are no longer in their native state, with the potential for specimen damage and artefacts being introduced during sample freezing. Liquid cell TEM not only preserves the liquid state of the specimens inside the TEM vacuum, but also allows the *in situ* observation of biological process [9].

*In situ* TEM has become a popular technique for observing reactions at the nanoscale [21]. Compared with other *in situ* techniques like atomic force microscopy (AFM), scanning tunnelling microscopy (STM) and various X-ray methods, TEM not only has the advantage of having both high temporal and spatial resolution, it also provides direct visualization of any changes in structural, morphological [22] or elemental distribution [23,24] at the nanoscale. The development of enclosed liquid cells has allowed the *in situ* TEM imaging of liquid phase reactions, opening up fields from electrochemistry to cell biology for study. This paper reviews *in situ* TEM experiments of liquid samples, with a focus on microchip encapsulated liquid cell TEM. We will cover the basics of the technique, its strengths and weaknesses with respect to related *in situ* TEM methods for characterizing liquid systems, and its application in biological, chemical and materials science fields.

## 2. Different TEM configurations for imaging liquid samples

In TEM, a high-energy electron beam is transmitted through a thin specimen, with the potential for elastic and inelastic scattering of some of the beam. The resultant beam can be collected and analysed in different ways, such as with high angle annular dark-field detectors (HAADF) to yield atomic number sensitive images, or the energy loss of inelastically scattered electrons can be measured to yield a characteristic spectrum (electron energy loss spectroscopy, EELS). There are two crucial requirements for this technique [25]. First, the TEM column needs to be maintained at a high vacuum, at least $10^{-5}$ Pa, to minimize undesired electron beam scattering and sample contamination. Secondly, specimens need to be thin enough for electrons to penetrate through, usually of the order of hundreds of nanometres for nanoscale imaging. These requirements are problematic for imaging liquid phase specimens; most liquids will simply vaporize under the high vacuum, and manipulating a liquid to remain sufficiently thin can be challenging. There are various approaches to tackling these issues, with the main methods for TEM imaging of liquid samples shown and compared in figure 1.

The most straightforward approach is to use low-vapour-pressure liquids where possible, since these liquids can withstand the high vacuum of the TEM [2,31–34]. These liquids can be imaged in the same way as solid specimens within TEM, with the ionic liquid simply applied to a standard specimen grid as shown in figure 1(i). Using micrometre-sized holes for wetting ensures that the ionic liquid droplets are sufficiently thin to allow for imaging. More complex methods have also been designed to perform *in situ* biasing experiments on ionic liquid specimens. For example, as shown in figure 1(ii), a miniature battery can be built and cycled *in situ* under TEM using an ionic liquid electrolyte, which provides the ionic conduction between the two electrodes. Upon applying electric potential, the charging/discharging processes can be performed as in a real battery, and the corresponding changes at the electrodes can be imaged using *in situ* TEM. The electrode material is in the form of nanowires to allow good image quality under TEM. This technique requires a special sample holder with electrical feedthroughs, so that a bias may be applied from outside the TEM. However, a major limitation is that this technique can only be used with low-vapour-pressure liquids. Most liquids cannot survive the high vacuum of the TEM, and they often cannot be replaced with a low-vapour-pressure alternative without fundamentally changing the nature of the reaction one plans to observe.

To address this, other approaches have been developed that can image liquids regardless of their vapour pressure. The 'open environmental cell', or environmental TEM (ETEM), maintains the specimen chamber at a relatively high pressure compared with the rest of the microscope by differential pumping. This approach has been widely adopted in SEM [35,36] to image liquid specimens. An ETEM is a special type of TEM that employs a series of pumps and apertures to gradually change the pressure along the column, as shown in figure 1(iii). Pressure as high as 2000 Pa can be achieved within its specimen chamber [28]. However, due to the limits of differential pumping, atmospheric pressures cannot be achieved using such a set-up, which means most liquids with high vapour pressure will still evaporate. This mean that common solvents like isopropanol, dimethyl carbonate, dichloromethane or even water cannot be easily used within such system. Any salt dissolved in them will precipitate out during experiment due to the rapid evaporation of the solvent. Although some liquid phase

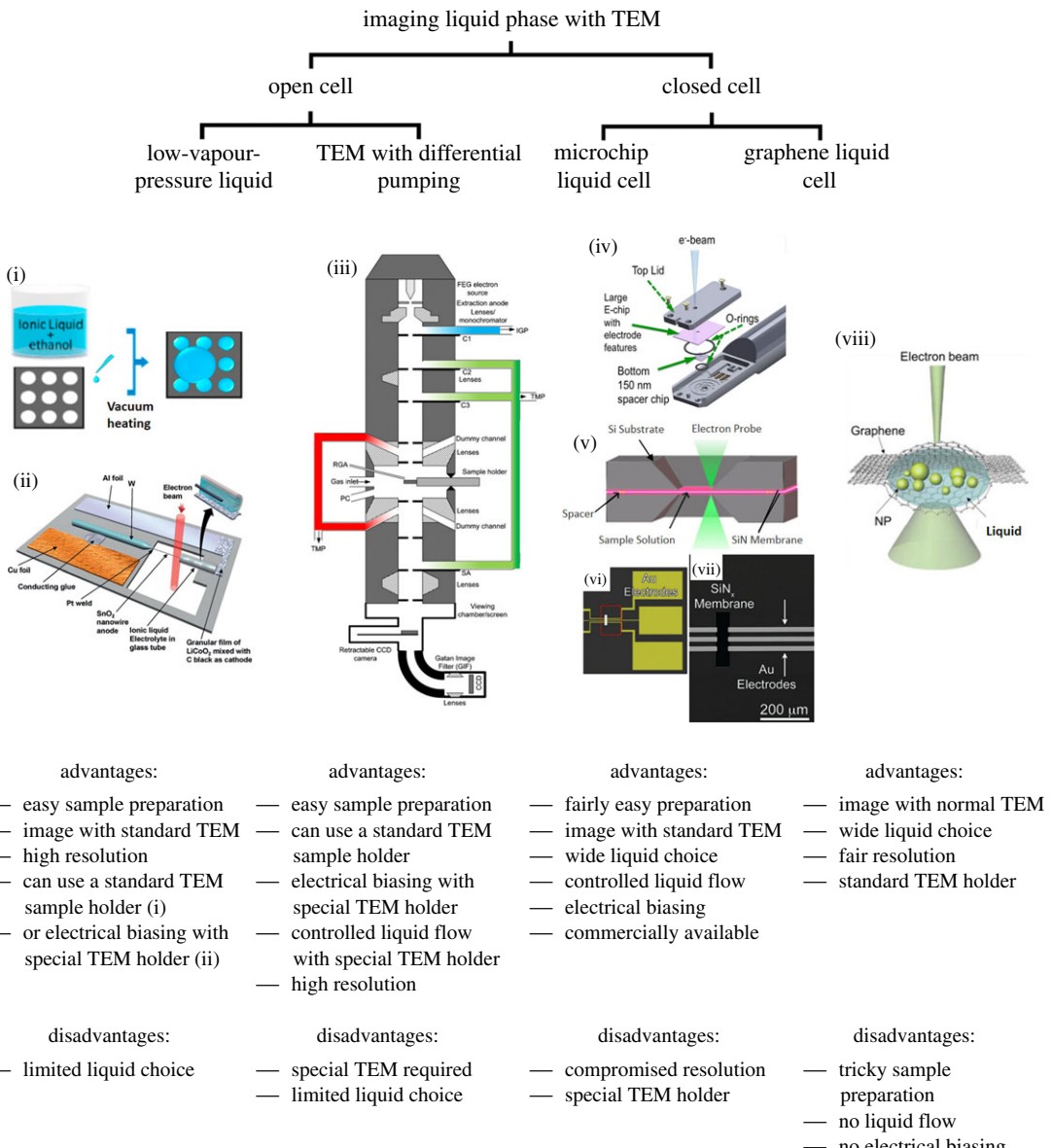

**Figure 1.** Comparison of different techniques for liquid phase TEM imaging. (i) Ionic liquid films maintained inside micrometre-sized holes from surface tension, suitable for TEM imaging [26]. (ii) Design schematic of a miniature battery that may be imaged inside TEM, using an ionic-liquid-based electrolyte [27]. (iii) Schematic of differentially pumped TEM column with three different pumping stages, indicated by different coloured pipes, permitting a high pressure sample chamber [28]. (iv) Schematic of an *in situ* liquid TEM stage on the tip of a TEM holder [29]. (v) Schematic of an assembled liquid cell from two silicon/silicon nitride chips [23]. (vi) Silicon chip with microfabricated electrodes [14]. (vii) Magnified SEM image of the red box region in (v), showing electrodes partially over the SiN$_x$ viewing window [14]. (viii) Schematic of a graphene liquid cell for TEM. The specimen is encapsulated between two graphene sheets [30].

experiments were successfully conducted within such systems [31,37], it is mainly used for gas phase experiments [38–42] and biological specimens [7,43], which do not require very high pressure to maintain.

Another approach can be defined as the 'closed environmental cell'. Unlike the 'open environmental cell' approach with differential pumps, the pressure of the closed cell is maintained by encapsulating the sample volume in screening membranes. Such a specimen can be maintained at a much higher pressure. The electron beam penetrates and images the specimen through robust and electron-transparent windows on the top and bottom of the encapsulated specimen cell. The cell can be in-built within the TEM specimen holder and is therefore compatible with conventional TEMs. A closed environmental cell can be used for both gas phase [44] and liquid phase imaging [1,3]. When used for liquid phase, it is usually called a 'liquid cell'.

A liquid cell is typically only hundreds of nanometres thick, confined by two thin but robust electron-transparent membranes. The commercially available and by-far most widely adopted liquid cell system is

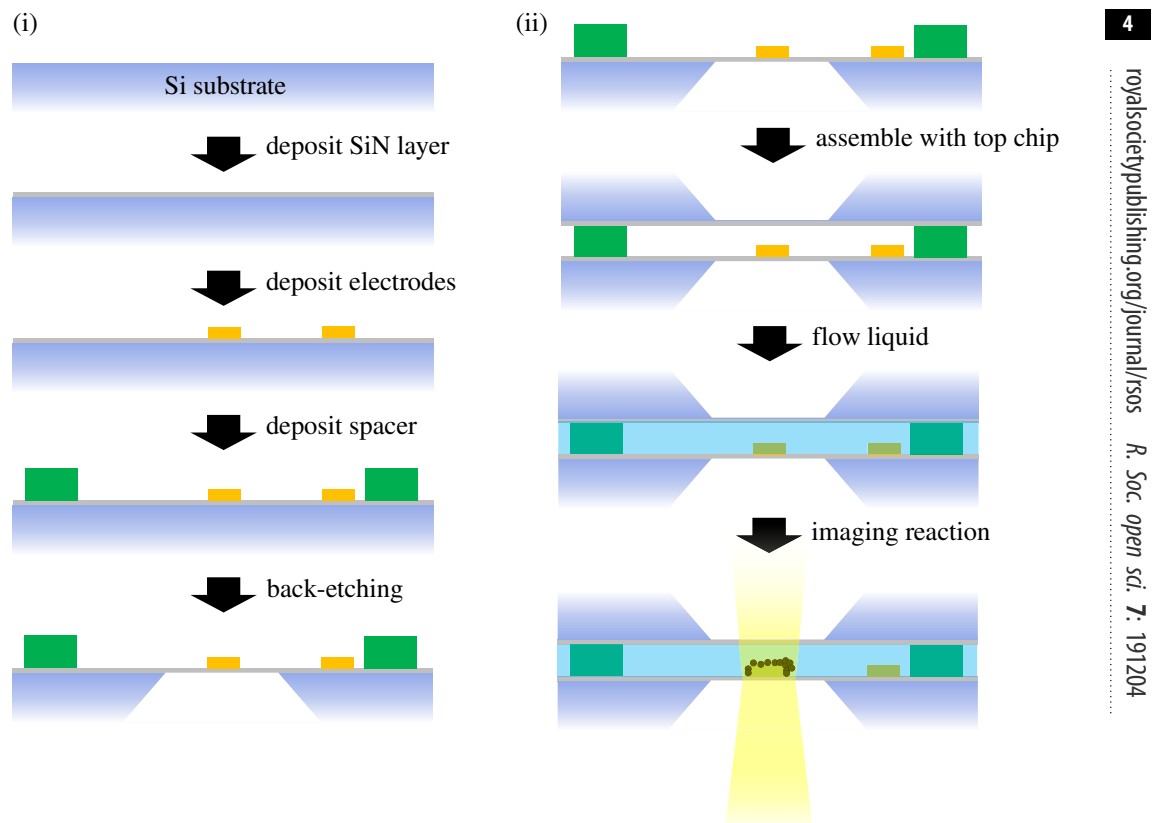

**Figure 2.** (i) Simplified schematics of the production process of a liquid cell microchip via photolithography. (ii) Schematics of the assembly of the microchips for liquid cell TEM imaging.

based on silicon microchips, as shown in figure 1(iv–vii). A thin layer of liquid phase specimen of between tens of nanometres to a micrometre can be confined between two microchips using O-rings and spacer materials. A thin layer of amorphous silicon nitride (SiN) is most commonly used as the membrane window material, through which the electron beam can penetrate and be used to image. (Detailed fabrication and application processes of this system is shown in the next section.) Such liquid cells can safely contain almost all types of liquid, independent of vapour pressure. In addition, microfabrication allows additional capabilities to be added, including allowing liquid flow [45], temperature control (heating and cooling [46]) and applying electrical bias using patterned electrodes.

A variant of the liquid cell is the 'graphene liquid cell', which has been used since 2012. Microencapsulation using graphene [10,47], or 3 nm thick amorphous carbon films [5], have been shown to well preserve volatile specimens like biological cells and liquid crystals within the ultra-high vacuum in TEM. This technique can also be used to image liquid specimens [48–50]. Compared with the microchip cell, the graphene cell can normally achieve better resolution due to less scattering from the window material and smaller liquid thicknesses. In addition, it does not require an expensive dedicated sample holder [51–55]. However, this technique is currently limited in several ways. First, encapsulation of a liquid sample with graphene requires very careful cell assembly due to the difficulties in handling graphene. Second, only a very small amount of liquid can be encapsulated and imaged, less than 0.01 picolitre, compared with the nanolitre capacity of a microchip liquid cell. Third, unlike with microchips, microfabrication cannot be used in this cell, which means that it is not compatible with functions like liquid circulation or applying electrical biasing. The lack of liquid flow, combined with the low volume of liquid between the graphene membranes, means studying dynamic processes is hindered by the limited availability of reactants. This review focuses on the microchip-based liquid cell, with a detailed discussion of the graphene liquid cell beyond the scope of this review.

## 3. Liquid cell design and application

Figure 2 illustrates the production and use of a typical liquid cell with electrodes. The liquid cell is usually fabricated by standard photolithography patterning of silicon microchips. A thin layer of SiN, typically

tens of nanometres thick, is first deposited on top of the Si substrate. Amorphous SiN is the most widely used electron-transparent membrane material in liquid cell TEM. Other membrane material alternatives include silicon oxide [56] and amorphous carbon [9]. Electrodes and spacers may then be deposited. The spacer is used to provide enough space between the two chips to ensure enough liquid remains in the cell during imaging and to allow for liquid flow. Electrodes can be employed to apply an electric potential to manipulate particle movement within the liquid or to induce electrochemical reactions during imaging. Finally, the Si is etched away from the back, leaving a small window of only the SiN membrane. The window is usually in a rectangular shape, as shown in figure 1 (vi,vii), with widths of tens of micrometres and lengths of over a hundred micrometres. While this limits the viewing area, wider windows are more susceptible to breaking and bowing, common issues associated with the liquid cell, which will be discussed in detail later. The electron beam is able to transmit through the windows to allow imaging. A comprehensive description of the production process of a liquid cell was given by Grogan & Bau [57].

The assembly of a liquid cell is shown in figure 2(ii). A second, flipped, chip is aligned on top of the bottom chip, normally fixed in place by clamping tight with O-rings and metal frames on the holder tip (figure 1(iv)), although wafer-bonding and epoxy have also been used by some researchers. The separation between the windows determines the thickness of the liquid phase specimen being imaged. Such separation is controlled by the height of the spacer being used. A thicker layer usually means better liquid flow and allows imaging of relatively large-sized specimens like biological cells [8,56]. However, a thick liquid layer leads to compromised resolution due to increased beam scattering [58]. The optimum resolution of the TEM image after scattering through a liquid layer is estimated to be [59]

$$d_{TEM} = \frac{A_L \cdot \alpha \cdot C_c \cdot T}{E^2},$$ 
(3.1)

where $d_{TEM}$ is the resolution, which is directly proportional to the objective semi-angle, $\alpha$, and to the liquid thickness $T$, and inversely proportional to the electron beam energy $E$. ($A$ is a constant that depends on the liquid and $C_c$ is the chromatic aberration coefficient.) For scanning-mode TEM (STEM) [58,59] the optimum resolution is also dependent on the liquid thickness:

$$d_{STEM} \propto T^{1/2}.$$ 
(3.2)

As we can see from both equations (3.1) and (3.2), thicker liquid means worse resolution. Another critical factor limiting the resolution is the dose rate. For both TEM and STEM, the resolution is dependent on the dose rate as follows [60]:

$$d_{TEM,STEM} \propto D^{-1/4}.$$ 
(3.3)

Dose rate should be carefully monitored and controlled during imaging to avoid beam damage, as will be discussed in detail later. A comprehensive study of all factors affecting liquid cell resolution in TEM and STEM was done by de Jonge *et al.* [60].

# 4. Liquid research with conventional TEM and open environmental TEM

This review is primarily focused on the TEM study of liquid samples that are encapsulated inside microchip liquid cells. However, for comparison we will briefly discuss some *in situ* TEM studies of open-cell liquid systems. Due to the chamber pressure limit of the conventional TEM and open ETEM, specimens need to withstand high vacuum during experiments if they are not encapsulated inside a cell. This limits the options of the liquid being imaged. Most works performed in an open system employ either a liquid phase alloy or an ionic liquid, both of which are particularly stable under high vacuum. Many of the pioneer works were performed on alloys heated close to their melting point [31–33]. These works closely examined the liquid–solid interface during melting and solidification, imaging processes like phase segregation, crystallization etc.

An interesting application of molten alloys is to serve as the catalyst during nanomaterial growth. With TEM, *in situ* growth of these materials can be captured as shown in figure 3a. In 2005, Ross *et al.* [2] showed the *in situ* growth of a Si nanowire in a disilane environment catalysed by a Au-Si alloy droplet at the wire tip at elevated temperature (500–650°C), which showed a faceting sawtooth structure. Both the width of the nanowire and the faceting period are highly dependent on the size of the Au-Si droplet. Based on these observations, it was later found that the Si nanowire growth can be precisely controlled via manipulating droplet geometry by applying an electric field [62]. This

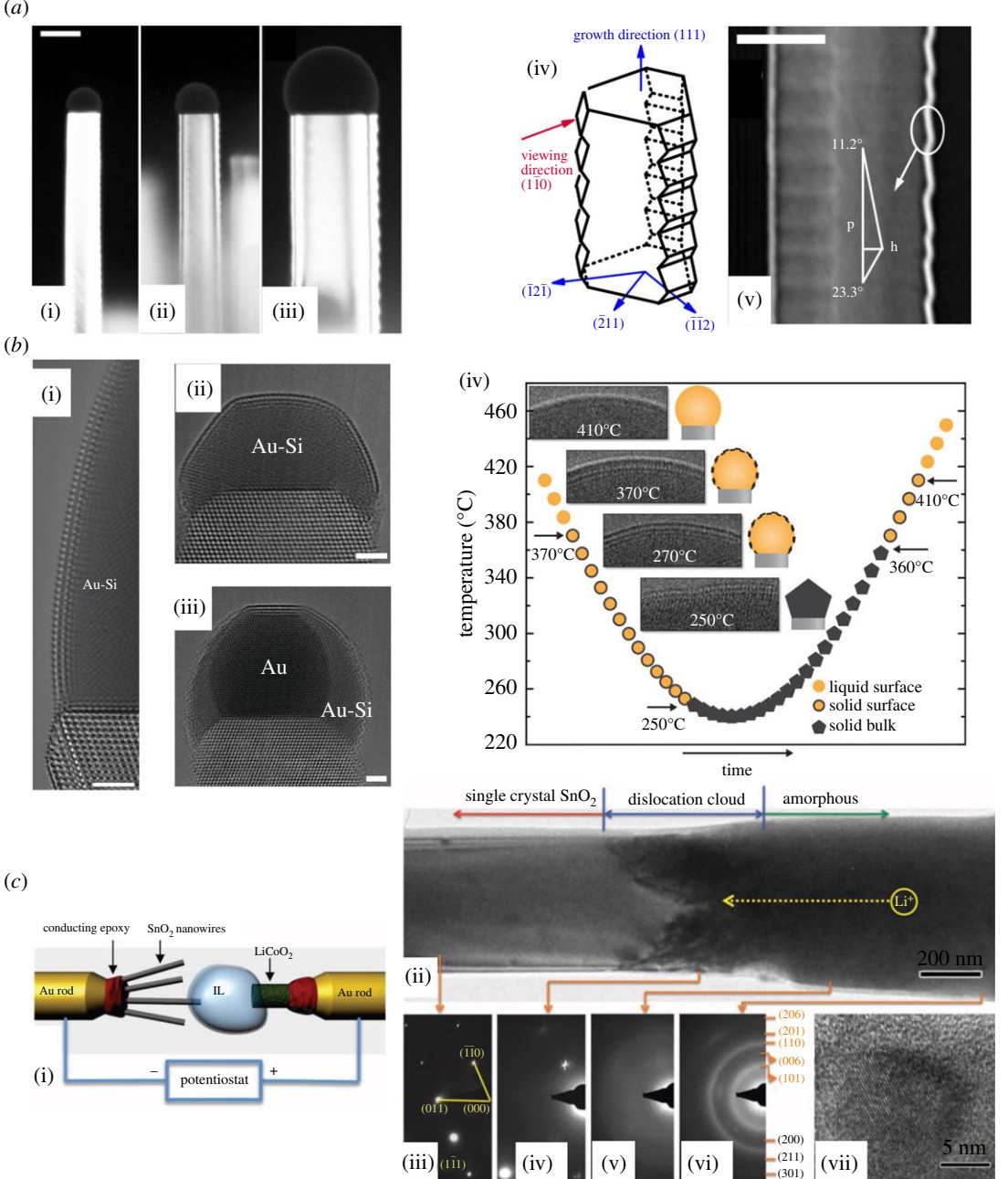

**Figure 3.** (a) Growth of saw-tooth faceting Si nanowires using liquid Au–Si droplet as the catalyst [2]. (i–iii) Three Si wires of different width grown from Au–Si droplets of three different sizes. Scale bar is 100 nm. (iv) Schematic of the three-dimensional structure of the nanowire. (v) The surface structure with facet angles indicated. p and h denote sawtooth period and amplitude. Scale bar is 50 nm. (b) Crystalline ordering at the surface of Au–Si liquid [61]. (i) A crystalline bilayer visualized on a 50 nm diameter droplet at 350℃. (ii) Surface ordering on a 12 nm diameter droplet at 350℃. (iii) A solid Au particle heated on Si at 360℃ where the eutectic liquid starts to cover its surface. Scale bars are 2 nm for (i–iii). (iv) The surface ordering evolution at different temperature. (c) Lithiation of a single SnO₂ nanowire anode at −3.5 V against a LiCoO₂ cathode [34]. (i) Schematic of the experimental set-up. (ii) The reacted ('amorphous') and non-reacted ('single-crystal SnO₂') wire separated by the lithiation interface 'dislocation cloud'. (iii–vi) Diffraction patterns taken from different sections of the nanowire in (ii) showing the amorphization upon lithiation. (vii) Sn nanoparticles dispersed in an amorphous matrix from the lithiated part of the wire.

illustrates how *in situ* TEM is able to provide unique insights into fundamental materials processes, which in this case was the relationship between the surface energy and morphology of the nanomaterial, and how to manipulate the growth of nanomaterials to achieve accurate and direct nanowire growth in devices [63]. Since then, this system was widely adopted to study *in situ* the three-phase growth (solid formed from gas, catalysed by a liquid alloy) of different nanowires [64–66].

A significant advantage of the open-cell system is the high resolution achievable due to the high vacuum and no interceding encapsulating layers. A good example of this is shown in figure 3b, which shows a stable atomically thin crystalline phase formed from liquid Au-Si and its changes with respect to different temperatures [61]. The imaged solid–liquid interface atomic structure is of interest to trace the individual atoms during solidification/melting to help better understand and model these processes.

Ionic liquids are liquid salts consisting of ions and ion pairs [67], which can dissolve many kinds of substances. They can withstand high vacuum and electron beam irradiation [68,69]. They are also good ionic conductors, which makes them perfect for use as alternative electrolytes for battery and other energy-related *in situ* TEM research. In 2010, the first experimental design for performing biasing under high vacuum with an ionic liquid in a TEM was published [27]. Within the same year, with this design, Huang *et al.* [34] studied the *in situ* lithiation of $SnO_2$ nanowires within TEM, as shown in figure 3c. (Lithiation/delithiation are the incorporation/removal process of lithium from an electrode in a lithium ion battery.) Upon lithiation, the crystalline $SnO_2$ nanowire was converted to amorphous $Li_2O$ with nanocrystalline Sn and $Li_xSn$ dispersed within it (as shown in figure 3c(vi)). The interface between lithiated and unlithiated wire contained a 'dislocation cloud', which significantly expanded and distorted the wire as it moved forward. This expansion and distortion was not reversible upon delithiation. This observation directly showed the degradation process of the electrode material and helps explain the loss in battery capacity due to cycling.

The same set-up can be used to test the charging/discharging effects within different battery systems [70] on a wide range of electrode materials [71–75] and to study the effect of coating [76,77] and doping [78] on their degradation processes. For example, Liu *et al.* [72] found that graphene nanoribbons showed greater flexibility and were highly resistant to fracture upon lithiation, unlike multi-wall carbon nanotubes [71] which always become brittle and prone to fracture after lithiation. The difference was ascribed to the unconfined stacking of graphene layers, which prevents the interlayer stress build-up and makes it potentially a highly durable electrode material. Visualization by TEM not only helps us to understand the degradation mechanisms within different battery systems, it can also help test different potential electrode materials, which can be beneficial in battery research to improve stability and capacity. However, using an ionic liquid as the test battery electrolyte means its representativeness toward real batteries is questionable, as they are not typically employed in batteries. A common battery electrolyte consists of a lithium salt, for example $LiPF_6$ or $LiClO_4$, dissolved in organic solvents, for example, dimethyl carbonate (DMC) or diethyl carbonate (DEC). These solvents have high vapour pressure [79] (DMC: 5300 Pa at 20°C; DEC 1400 Pa at 25°C), which cannot be maintained in an open-cell configuration. A closed cell is required to image battery systems with these more relevant electrolytes.

# 5. Research with liquid cell

## 5.1. Nanomaterials research

### 5.1.1. *In situ* growth of single phase nanoparticles

The microchip-based closed liquid cell has a wide range of applications, due to the ability to pattern electrodes and liquid flow spacers on them, and their robustness towards a wide variety of liquids. So far, the most common application is to study the synthesis and manipulation of nanomaterials. Table 1 shows a selected list of liquid cell experiments on nanomaterials with significant importance.

The first *in situ* liquid cell TEM nanomaterial synthesis was performed by Williamson *et al.* in 2003 [1]. This work visualized the *in situ* electrochemical nucleation and growth of Cu clusters. The same experiment was later repeated in 2006 [97–99] with better resolution, which allowed its growth mechanism to be modelled and quantified. This is shown in figure 4a; a negative potential was applied from +0.3 V to −0.27 V with respect to a Cu reference electrode. Then a positive potential was applied back to +0.3 V. The deposition was a two-stage process. The first stage started immediately below 0 V potential. During this stage, the growth was reaction limited (lowering potential gives higher deposition rate due to higher driving force). Towards −0.1 V, the second stage starts, where the deposition became mainly kinetically limited (deposition rate starts to decrease upon further lowering potential). Upon applying positive potential, all the deposited Cu was stripped away. By measuring the size change of the Cu particle with TEM, such two-stage deposition was for the first time visualized and well correlated with the electrochemical I–V curve.

**Table 1.** Selected list of liquid cell experiments on nanomaterials performed since 2003.

| materials | author | methods | observation | importance |
|---|---|---|---|---|
| Cu cluster | Williamson et al. 2003 [1] | biasing via electrodes | nucleation, growth | first liquid cell experiment |
| Pt nanoparticle | Zheng et al. 2009 [4] | E beam illumination | nucleation, growth | sub-nm resolution allows clear visualization of growth mechanism |
| PbS nanoparticle | Evans et al. 2011 [6] | E beam illumination | nucleation, growth | different morphology from different solution; first atomic resolution in liquid cell |
| Ag nanoparticle | Woehl et al. 2012 [80] | E beam illumination | nucleation, growth | different morphology formed from different beam current. |
| iron oxyhydroxide nanoparticles | Li et al. 2012 [81] | premade | oriented attachment | in situ crystal movement and attachment with atomic resolution |
| Pt₃Fe nanorod | Liao et al. 2012 [82] | E beam illumination | nucleation, growth and assembly | first in situ nanorod formation |
| Au–Pd core–shell nanoparticle | Jungjohann et al. 2013 [83] | E beam illumination | growth of the shell onto the core | first in situ core–shell synthesis by coating the core with shell material |
| Pt nanoparticle | Liao et al. 2014 [84] | E beam illumination | nucleation, growth | first in situ crystal growth with atomic resolution |
| Pd nanoparticle | Jiang et al. 2014 [85] | oxidative etching by E beam | dissolution | first in situ oxidative etching of nanoparticle |
| Ag–Pd nanocage | Sutter et al. 2014 [86] | galvanic replacement of Ag using a Pd solution | formation | first in situ formation of nanocage |
| Au nanoparticle | Hermannsdörfer et al. 2015 [87] | premade particles change pH, NaCl concentration, beam dose | dissolve, merge | in situ change of particle with respect to environment |
| Au nanoparticle | Park et al. 2015 [88] | E beam illumination | nucleation, growth | quantify the relationship between beam dose with growth rate |
| Au nanoparticle | Alloyeau et al. 2015 [89] | E beam illumination | nucleation, growth | |
| Fe₃Pt–Fe₂O₃ core–shell nanoparticle | Liang et al. 2015 [90] | E beam illumination | nucleation and growth | first in situ growth of the complete core–shell structure |
| Pt–Au core–shell nanoparticle | Wu et al. 2015 [91] | E beam illumination | growth of the shell onto the core | quantify the growth kinetics of core–shell structure, showing the three different stages during growth |

(Continued.)

**Table 1.** (Continued.)

| materials | observation | author | methods | importance |
|---|---|---|---|---|
| UiO-66(Zr) & ZIF-8 MOF | nucleation, growth and assembly | Patterson et al. 2015 [92] | mixing solutions and heating | first liquid cell in situ experiment on MOF |
| Pd nanoparticle | nucleation, growth, size control | Abellan et al. 2016 [93] | E beam illumination | precise size control growth to sub-3 nm using capping agent |
| Ag nanoparticle | nucleation, growth | Ge et al. 2017 [94] | E beam illumination | catalytic growth due to the presence of Pt |
| Ag particle capped with ligands | oriented attachment | Zhu et al. 2018 [95] | premade | the role of capping ligands on the oriented attachment process |
| Pd nanoparticle | nucleation, growth | Yang et al. 2019 [96] | electrochemical biasing | significant effect of HCl on the growth mechanism and resultant morphology |

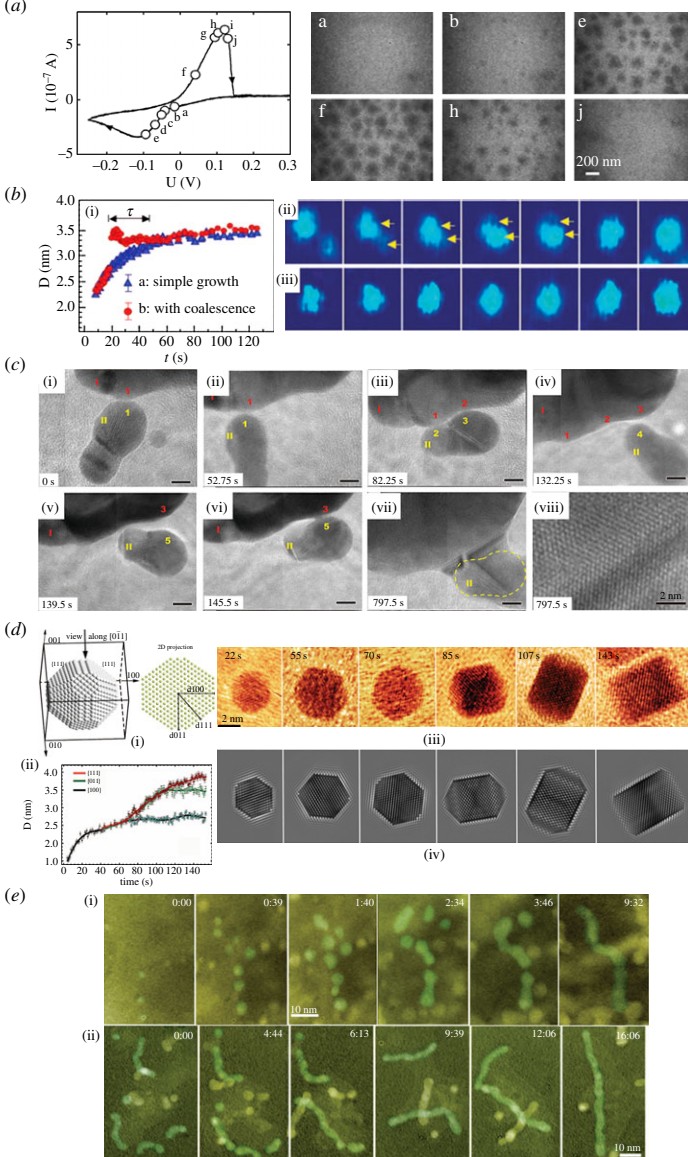

**Figure 4.** (*a*) Images recorded during the electrochemical deposition and stripping process of copper; a, b, e were recorded during deposition; f, h, j were recorded during stripping [97]. The cyclic voltammogram on the left-hand side shows the point in the cycle at which each image was recorded. (*b*) Platinum nanocrystal growth trajectories. (i) Particle size versus growth time for these particles with two different types of growth trajectories. (ii) Coalescence growth. (iii) Monomer addition growth (simple growth) [3]. (*c*) Direction-specific oriented attachment of iron oxyhydroxide nanoparticles. (i–vii) Sequential dynamics of the attachment process of two particles (particle I and II), the surfaces of particles I and II made transient contact at many points and orientations (1–1, 1–2, 2–3 and 3–4) before final attachment and growing together (point 3–5), scale bar 5 nm. (viii) Attachment interface from (vii), showing an inclined twin plane [81]. (*d*) *In situ* facet development of a Pt nanoparticle. (i) The atomic model of a Pt nanoparticle and its projection along the [011] zone axis. (ii) Average growth profile along three different directions (100), (011) and (111). (iii) Sequential growth of the Pt nanoparticle. (iv) Simulated TEM images of the Pt nanoparticle in (iii). (*e*) [82]: Real-time imaging of $Pt_3Fe$ nanorod formation. (i) Nucleation and growth of $Pt_3Fe$ particles followed by their assembly into a nanorod. Particles contributed to the nanorod are highlighted in green. (ii) The growth of a long $Pt_3Fe$ nanorod from the assembly of several short nanorods [84].

By improving the liquid cell design and using intensive electron beam radiation instead as the means to instigate nucleation and growth of nanoparticles, Zheng *et al.* [3] managed to improve the resolution of the liquid cell from 5 nm to sub-nanometre. A high electron beam dose can lead to radiolysis of the liquid, yielding radiolysis species like hydrated electrons, $e_h^-$ [100], which act as a strong reducing agent and thus cause the reduction of the solution and the formation of the nanoparticles. Using this technique, the nucleation and growth of nanosized Pt particles was clearly visualized. With this high-resolution

technique, Zheng *et al.* revealed two different growth mechanisms during the early stage of nanocrystal growth (coalescence versus monomer attachment), as shown in figure 4*b*. In addition, by tracing the growth rate of individual particles, it was found that a monodisperse size distribution could always be reached regardless of the initial growth path, as shown in figure 4*b*(i). Zheng *et al.*'s method of instigating nanomaterials growth by beam illumination has been widely adopted to study nanomaterials synthesis with liquid cell TEM. For example, using such method, Evans *et al.* [6] grew PbS nanoparticles of different morphologies with precursor solutions of different chemical compositions. In the same paper, Evans *et al.*, for the first time, showed that lattice resolution could be achieved within liquid cell TEM.

Apart from nanoparticle nucleation and growth, many other processes can also be visualized *in situ* with atomic resolution. For example, in 2012, the oriented attachment process of iron oxyhydroxide nanoparticles was visualized by Li *et al.* [81], as shown in figure 4*c*. The nanocrystal kept moving and rotating around another bigger particle until a point where lattice matching with the adjacent nanocrystal was achieved. Then they merged into a single crystal. Such oriented attachment process was later also visualized in Au nanoparticles capped with organic ligands by Zhu *et al.* [95], who pointed out the importance of ligands in this case. Ligands guide the rotation of the particles to share a common {111} orientation. {111} has the lower ligand binding energy and thus it is the preferential orientation for attachment.

In 2014, the first *in situ* nanocrystal growth with atomic resolution was performed by Liao *et al.* [84], as shown in figure 4*d*. With such high resolution, the growth rate along different crystallographic directions of a Pt nanocube could be individually tracked. Such visualization greatly enhanced the understanding of crystal growth with respect to surface energy of each facet, which is the key to the manipulation of nanocrystal growth. And apart from synthesis, the dissolution process of different nanoparticles has also been studied using liquid cell TEM [85,101,102]. For example, Wu *et al.* [102] visualized the *in situ* dissolution of platinum nanoparticles, from which a quantitative kinetic model was developed, taking into account different crystal structures and the atoms' location within them.

### 5.1.2. The growth of more complex structures

Apart from single-phase nanoparticles, liquid cell TEM has also been used to image the growth of more complex systems. For example, much research has been done on core–shell structured nanoparticle synthesis. The first liquid cell experiment on a core–shell structure was performed in 2013 by Jungjohann *et al.* [83], who synthesized the Au-Pd core–shell structure by coating Au nanoparticles with Pd. It was shown that a different core morphology could lead to different shell growth. In 2015, Liang *et al.* [90] demonstrated the *in situ* growth of $Fe_3Pt$-$Fe_2O_3$ core–shell structure, during which both the core and the shell were formed by the action of the electron beam, as described in the previous section. The depletion of Pt precursor in the solution terminated the core growth and further electron beam illumination led to the epitaxial growth of the shell. With the same technique, Zheng *et al.* [103] demonstrated the *in situ* growth of the PtNi-Ni core–shell structure, showing the feasibility of applying a metallic shell as a potential strategy to protect novel catalyst materials during reactions. The growth kinetics of core–shell structures was studied and quantified by Wu *et al.* [91]. Using the Pt-Au as the model system, three different growth stages of the Au shell were revealed; firstly, Au deposited on the corner sites of Pt icosahedral nanoparticles, then Au diffused from corners to terraces and edges, and finally Au grew on Au surfaces layer-by-layer to form the final Pt-Au core–shell structure.

The formation of nanorods from the assembly of nanoparticles was demonstrated by Liao *et al.* [82]. As shown in figure 4*e*. $Pt_3Fe$ nanoparticles were nucleated and grown with the electron beam. A uniform size distribution was reached and stabilized by the surfactant. A self-assembly process of the particles (attachment, reorientation and size adjustment) then led to the formation of a single crystal nanorod. The effect of surfactant concentration [104] on the morphology and stability of the formed $Pt_3Fe$ nanocrystals and nanorods were also studied.

Other *in situ* liquid cell TEM works on more complex systems include nanocage formation using galvanic replacement [86], metal–organic framework's (MOF) nucleation and growth rate with respect to different metal-to-ligand ratio [92], iron Keggin ion's building/conversion process to magnetite and ferrihydrite [105], the growth of various metal–Fe–oxide nanoparticles using various combinations of different precursor metal solutions [106] etc. These studies demonstrate the power of liquid cell TEM to image a wide range of nanomaterial structures, which can help us understand their growth mechanisms, and thus allow us to make informed choices in our synthesis strategies for nanomaterial preparation. With such knowledge, better control of the growth and tailoring of the morphology can be achieved.

### 5.1.3. The effects of the growth environment

As shown before, applying bias and electron beam irradiation are the most common ways to grow nanostructures in liquid cell TEM, although temperature control can also be used to induce growth and dissolution [107]. Regardless of the growth method, two key factors always dominate the growth kinetics and the resultant material: precursor solution composition and electron beam radiation. Many works have been performed to assess their effects.

Different precursor solutions can lead to different growth mechanisms. For example, Yang *et al.* [96] showed that by adding HCl into the precursor solution, the Pd particles switched from three-dimensional island growth to aggregative growth. Such visualization helped explain the change in the resultant morphology (from smooth to porous and open morphology). Abellan *et al.* [93] visualized how, by adding a capping agent tri-n-octylphosphine (TOP) to the solution, precise size control of the grown Pd nanoparticle to sub-3 nm can be achieved. Ge *et al.* [94] showed anomalously fast growth rates of Ag nanoparticles when the growth took place adjacent to Pt, providing experimental proof of the catalytic effect of Pt on the growth of nanomaterials. A future avenue for such *in situ* TEM experiments could be to use the liquid flow lines into the liquid cell to inject different reactants at sequential stages, allowing the controlled growth of more complex nanostructures. Apart from precursor solutions, using different electron beam currents can also lead to different growth mechanisms, and thus resultant morphologies, during nanoparticle growth in a liquid cell, as shown by Woehl *et al.* [80]. Systematic studies to quantify the beam current effect were later performed by Park *et al.* [88] and Alloyeau *et al.* [89] using Au nanoparticles as the model system. Park *et al.* showed that a minimum dose rate is required to nucleate the Au nanomaterials due to the competition between the reduction and oxidation processes in the solution. Above such limit, particles' growth rate follows a power law with dose rate. However, such power law relation does not apply to very high electron beam dose, as shown by Alloyeau *et al.* [89]. Alloyeau *et al.* showed that, at relatively lower electron dose, the growth is thermodynamically driven, i.e. a higher growth rate under a higher electron beam dose. However, an upper limit exists, from which the growth becomes dominated by kinetics, i.e. diffusion, and higher dose cannot further increase growth rate. The morphology of the grown particle is also shown to be different for thermodynamically dominated and kinetically dominated growth.

These *in situ* experiments allow us to better understand and model growth mechanisms within different environments, which is useful to grow, stabilize and manipulate different nanostructures. For example, Zečević *et al.* [108] showed that under STEM, the narrowly focused electron beam probe can elongate silica nanoparticles. Tian *et al.* [109] showed that the electron beam can split AgCl nanocrystals, which reassemble when illumination terminates. As we will discuss later, often the electron beam effect is something that the experimentalist wants to avoid, with beam-induced damage changing the behaviour of the sample in adverse ways. However, these above experiments show that the beam can be valuable as the trigger to instigate reactions inside the TEM.

*In situ* imaging can also be used to test materials for different applications. For example, Hermannsdörfer *et al.* [87] performed a comprehensive study of the environment's effect (pH, NaCl, beam dose) on Au nanoparticles' behaviour (dissolution or merging), showing the change in particle size and morphology in response to the change in environment. Au nanoparticles can be potentially used for medicine, like imaging and protein targeting drug delivery. Such *in situ* work is useful since it is important to know how Au particles behave in different parts of the body.

### 5.1.4. Nanoparticle movement and interactions within the liquid cell

Liquid cell TEM can be used to image the movement and interactions of nanoparticles directly at nanoscale resolutions, rather than relying on the indirect measurements of spectroscopy techniques. Zheng *et al.* [4] showed the *in situ* observation of Au nanoparticle and nanorod motion in a liquid thin film driven by solvent evaporation. Three distinct modes of motion of nanorods were observed: when the particles are relatively far from the fluid front, its motion is a combination of the slow displacement mode (Brownian motion) and the occasional jump mode (nanorod rolling). The fastest motion mode took place when the nanorods were close to the drying patch where they were dragged by the fast-moving fluid front. However, solvent evaporation is hard to control within a liquid cell. Thanks to the development of liquid cell with controlled liquid flow [45], nanoparticle movement with respect to liquid flow can be systematically studied with respect to the flow rate. For example, Verch *et al.* [110] demonstrated that the Au nanoparticles' motion close to the membrane is three orders of magnitude slower than expected given the flow rate. By quantitatively comparing the nanoparticle motion with respect to electron dose,

particle coating, membrane surface charge, viscosity and liquid thickness, it was found that such slowing is due to the formation of an ordered layer with viscosity five orders of magnitude larger than bulk liquid due to the surface charge of the silicon nitride windows.

Particle motions due to various interactions can also be studied using liquid cell, including nanoparticle–nanoparticle interactions [111–113], particle–beam interactions [114], particle–bubble interaction [115] etc. For example, Park et al. [113] showed how Au nanoparticles may self-assemble into a superlattice. By tracking the movement of individual nanoparticles with respect to time, he was able to quantify the driving force for the process [116]: strong long-range anisotropic forces first drive the formation of chains from particles, then these chains fold to form the closed-pack superlattice due to close-range van der Waals forces. Using the same technique, processes like nanomaterial growth by particle attachment [81,117] can also be tracked and better understood.

## 5.2. Battery research

### 5.2.1. Dendrite formation

Metallic dendrite formation within a battery can lead to short-circuiting and even explosion. Upon cycling, Lithium metal can deposit and form dendrites on the anode. These dendrites can pierce through the battery separator and establish an electronic connection between the two electrodes. Such short-circuiting can lead to excessive localized heating which can lead to explosion. Further, the irregular deposition of dendritic lithium can lead to the build-up of 'dead lithium' that is not removed. This reduces the efficiency and capacity of the battery. Understanding this dendrite formation mechanism is of paramount importance to allow the application of more efficient and safer batteries with lower capacity loss upon cycling.

Since Li is a very light and reactive element, its in situ dendrite formation experiment was first difficult to conduct under TEM. Instead, most early in situ TEM dendrite growth experiments with liquid cells were performed with other metals, including Au, Pd [118], Cu [119] and Pb [120,121]. These experiments improved the understanding of metallic dendrite growth in general, based on which the in situ study of the Li system was later realized. The work on Pb dendrite by White et al. [120] is particularly interesting, as shown in figure 5a. Not only were the plating and stripping process of Pb dendrites clearly revealed, the change in $Pb^{2+}$ ionic concentration was also visualized by tracing the change in image contrast after subtracting two consecutive frames from the in situ movie. Such visualization is useful since it can trace the ion diffusion process, which is critical to the understanding of the growth kinetics of the dendrite. With better spatial and temporal resolution, the growth kinetics can then be quantified.

The visualization of Li dendrite formation was realized in 2014, during which a few researchers, Zeng et al. [15], Mehdi et al. [29] and Sacci et al. [14], managed to capture the Li dendrite formation and dissolution with liquid cells using common commercialized Li battery electrolytes like $LiPF_6$/EC/DEC and $LiPF_6$/PC. All three works reported the formation of a solid–electrolyte interface (SEI), which is a critical factor in battery research, dominating the plating and stripping processes during battery cycles. Yet the observed SEI by the three groups of researchers are very different in each case: Zeng et al. showed a uniform layered SEI formed on the Au electrode first, and then Li dendrite deposited on top of the SEI layer. Sacci et al. showed that the SEI, rather than a uniform layer, formed in a dendritic morphology, after which Li deposited in between these SEI dendrites. Mehdi et al.'s results can be seen in figure 5b,c. Figure 5c shows a uniform layered SEI formed on top of the deposited Li. However, such SEI only formed on Li dendrites that were still in good electric contact with Pt electrode. 'Dead' Li no longer in contract with the Pt did not form such SEI, and could not be stripped during delithiation. These 'dead' Li parts become detached from the electrode during delithiation, shown as the residual pieces after 'Li dissolution' in figure 5b. Despite the difference in the reported SEI morphology and location, one thing in common in all three reports was that the SEI does not change much during the stripping process. Another observation reported by Mehdi et al. was that cracking could be found in the Pt electrode after cycling, which can be seen in figure 5b. Such cracking is probably due to the strain from the Pt-Li alloying process due to lithiation. Such cracking was also reported in Au electrode by Zeng et al. [124].

Additives are commonly introduced into electrolytes in commercial battery to stabilize SEI formation and to supress dendrite formation [125]. Using Zn deposition as a model, Park et al. [126] successfully showed that the addition of Bi into the electrolyte can greatly mitigate the formation of Zn dendrites and gives a much more uniform and smooth coverage over the electrode. The additive effect on Li dendrite formation was studied in situ in 2015 by Mehdi et al. [122,127]. By adding trace amount of water, HF was formed and studied as an additive. This is shown in figure 5d. With more HF as the

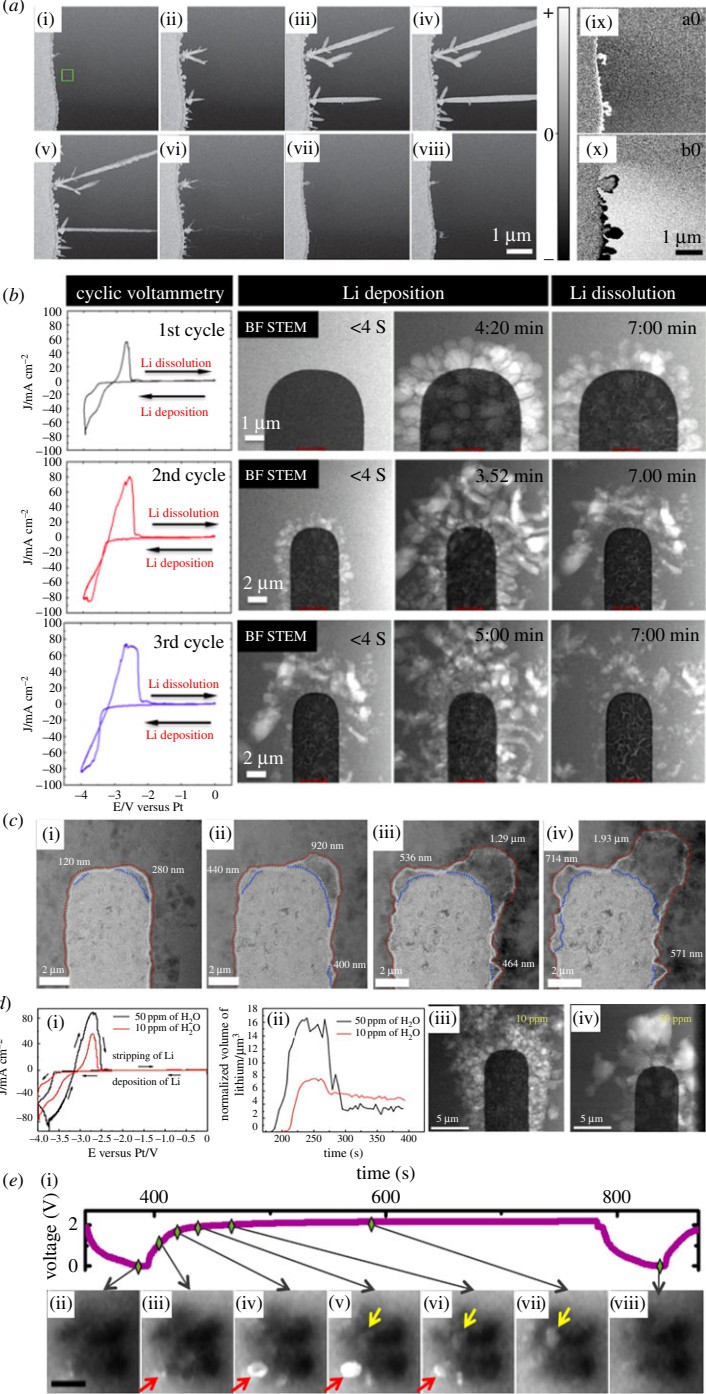

**Figure 5.** (*a*) *In situ* study of lead dendrite plating and stripping from lead ions in aqueous solution [120]. Sequential dendrite plating at −1.3 V (i–iv) and stripping at +0.3 V (iv–vii). (viii) Specimen surface after two cycles of plating and stripping. (ix–x) The change in Pb$^{2+}$ concentration in the liquid visualized by subtracting two consecutive frames from the *in situ* movie, lighter or darker regions indicate where the intensity increases or decreases. (*b*) *In situ* Li plating and stripping on Pt working electrode in LiPF$_6$/PC [29]. The first, second and third rows correspond to the first, second and third charge/discharge cycles, showing the cyclic voltammogram, electrode before plating, after plating and after stripping for each cycle. (*c*) The interface between the Pt working electrode and the LiPF$_6$/PC electrolyte after cycles 5–8, the red line shows the outside of the bright contrast SEI layer while the blue line shows the outline of the Pt electrode [29]. (*d*) Effect of HF formed from water as an additive to the Li plating and stripping process in LiPF$_6$/PC electrolyte [122]. (i) Cyclic voltammograms of electrolyte with 10 and 50 ppm water. (ii) Quantified total area of Li deposited and stripped for the electrolyte with 10 and 50 ppm of water. (iii) Li deposited on Pt working electrode in electrolyte with 10 ppm water. (iv) Li deposited on Pt working electrode in electrolyte with 50 ppm water. (*e*) Sequential evolution of a LiFePO$_4$/FePO$_4$ cluster during one cycle [123]. (i) The charge/discharge voltage profile. (ii–viii) 5 eV EFTEM images, brighter particles indicated by the arrows are FePO$_4$ particles formed from delithiation of LiFePO$_4$ during charging. Scale bar 200 nm.

additive, rather than the typical dendrite morphology, a smooth and dense layer was deposited. In addition, more Li was deposited and stripped during the cycle and a better coulombic efficiency was achieved. This is due to the increased LiF concentration within the SEI, which helped sustain faster $Li^+$ cation diffusion through the highly conductive LiF channels. The effect of $LiNO_3$ as an additive was also studied [128], which has been shown to induce distinctive difference on the dendrite morphology.

Other battery systems beyond Li-ion can also be tested with liquid cell TEM. Mg plating and SEI formation were visualized by Singh *et al.* [129] in 2018. The non-dendrite growth and the formation of the highly functional SEI, which allows continuous deposition and dissolution of Mg metal, have shown Mg as a promising candidate for future batteries. Liquid TEM provides us a means for the direct visualization of time-resolved dendrite morphology change [130], allowing such processes to be quantified [131] and for the growth mechanism to be better understood. This can help us find good ways for morphology control to mitigate the dendrite problem in batteries, via tuning of the electrolyte and electrode chemistry and electrode topography.

### 5.2.2. Electrode materials lithiation/delithiation

As mentioned, lithiation/delithiation are the incorporation/removal process of lithium from an electrode in a lithium ion battery. The volume and structural change during lithiation and delithiation is one of the key electrode degradation mechanisms which leads to battery capacity loss and failure. Therefore, understanding these processes within electrode materials is critical.

Lithiation has been performed on various materials with ionic liquids using open-cell TEM. However, ionic liquid is rarely used in actual batteries so the representativeness of such experiments compared with real lithiation processes in batteries is limited. The development of the TEM liquid cell has allowed lithiation experiments to be performed using commonly used Li-battery electrolytes. Gu *et al.* [132] showed the *in situ* lithiation/delithiation process of Si nanowire within a liquid cell. This experiment showed different dynamic structural evolution to the case for the open cell. This set-up allowed more insight into understanding of the lithiation process in common electrolytes, for example, the formation and evolution of the SEI.

Another way of studying electrode materials is depositing these materials onto the Pt or Au electrodes of commercialized liquid cells, as performed by Holtz *et al.* [123], and shown in figure 5*e*. $LiFePO_4$, one of the most commonly used cathode materials, was deposited onto the cell electrode using an inkjet printer capable of precise localized deposition of picolitres of $LiFePO_4$ nanoparticle suspension. By using energy-filtered TEM (EFTEM), the transition between the lithiated $LiFePO_4$ state and the delithiated $FePO_4$ state can be clearly distinguished during charging and discharging. In 2015, Zeng *et al.* [133] performed lithiation and delithiation of $MoS_2$ nanosheets, showing the irreversible decomposition process of nanosheets into nanoparticles upon lithiation. By using commonly used commercialized electrolyte, these experiments are more representative of the actual degradation process of the electrodes in commonly used Li batteries. Other than batteries, the liquid cell can also be used to test other energy materials, including for fuel cells [134] and photocatalysts [135].

One aspect that is important to control for in these experiments is the effect of the electron beam on the electrolyte. As mentioned, the electron beam can lead to the formation of radiolysis species like hydrated electrons, $e_h^-$, a reducing agent, which may lead to the degradation of electrolyte, depositing various species without electrical biasing. Abellan *et al.* [136] tested the electron beam's effect on several common Li electrolytes, and showed strong beam-induced artefacts especially for $LiAsF_6$ solutions. $LiPF_6$ salt electrolytes were found to exhibit fewer artefacts under the beam. For *in situ* TEM results to be representative, it is always a good idea to check that such beam effects are not too severe by performing control experiments.

## 5.3. Biological cells

Applying *in situ* liquid cell TEM toward the study of biological specimens has emerged as an alternative to the more commonly used cryogenic TEM technique. In 2008, Liu *et al.* [56] demonstrated that a closed cell can be used to image biological specimens. Two types of bacteria and their *in situ* tellurite reduction process were imaged. Liu *et al.* also showed that neither the sealing nor the TEM beam does significant damage to the studied bacteria. Within the same year, de Jonge *et al.* [8] demonstrated for the first time that a liquid cell can be used to image a whole eukaryote cell (fibroblast) of several micrometre size fully immersed in liquid.

Visualizing cells without significant damage requires imaging with a low dose. Combining this limitation with the presence of a thick liquid layer makes it difficult to achieve reasonable resolution

or contrast [137]. However, de Jonge showed that labelling with Au-tagged epidermal growth factor (EGF) molecules, a resolution down to 4 nm could be achieved despite having a liquid thickness of $7 \pm 1$ µm. Since then, combining Au-tagging and phase contrast in STEM mode became a popular technique to identify and trace the *in situ* changes of specific proteins within live biological cells in liquid state [138]. For example, in 2011, using such a technique, Peckys *et al.* [11] were able to identify nanoparticles around vesicles which helped trace and quantify vesicles' accumulation process.

Another advantage of using liquid cell for biological specimen imaging is that cells can be easily grown onto the SiN membrane. And by labelling certain proteins using both affibody and quantum dots, the same cell can be imaged with light microscopy [139], SEM and STEM [140] using a technique called correlative fluorescence and electron microscopy [139,141,142]. This is demonstrated in figure 6. HER2 within a SKBR3 cell were labelled with quantum dots, which can emit bright fluorescence signals under conventional light microscopy. Quantum dots' electron-dense core can also be easily distinguished under STEM. In this case, the STEM was performed with an environmental SEM with saturated water vapour atmosphere. However, the same chip is also compatible with imaging under liquid cell STEM, as shown in figure 6b(iii).

This labelling technique, combined with high-resolution TEM, can be used for quantitative study of the distribution of the proteins within the cell [143]. For example, from the inter-label distances, single-labelled protein monomer and double-labelled protein pairs can be distinguished, and from their distribution, Peckys *et al.* [144] revealed the dimeric conformation of the TMEM16A protein. This technique is an efficient way to elucidate the stoichiometry distribution and any possible change of various proteins on the cell membrane [47,145,146]. And not only proteins, it has been shown that nanoparticle labelling can also be done on DNA [49].

Apart from imaging proteins, *in situ* interaction studies of cells can also be performed. For example, Pohlmann *et al.* [147] monitored the interaction of glioblastoma (brain tumour) stem cells with nanoparticles. Such visualization can help the understanding and improvement of nanoparticle-based therapy to treat diseases like glioblastoma.

## 5.4. Other research works based on liquid cell TEM

Apart from nanomaterial synthesis and manipulation, battery cycling and biological cells, liquid cell TEM has also been used in a wide range of other fields. Examples include localized corrosion kinetics [16,148,149], biological mineralization processes [150–152], and *in situ* patterning [153] and electron beam lithography [154]. And with better resolution and more customized functionalities, more research fields will be explored using liquid cell TEM in the future [155].

# 6. Challenges and possible future improvements

There are three main challenges in performing liquid cell TEM: resolution, electron beam damage and representativeness.

## 6.1. Resolution

Liquid cell TEM suffers from poor resolution mainly due to electron scattering from the SiN window and the liquid layer. Liquid thicknesses of up to several micrometres can be imaged, but resolution will be heavily compromised [8,11]. A quantitative study showing the correlation between the resolution and the liquid layer thickness was done by de Jonge *et al.* [58].

Unnecessarily thick liquid can be formed due to bowing of the membrane, leading to reduced resolution [11,57]. Such bowing is caused by the pressure difference between the inside and outside the liquid cell. There are several possible ways to avoid bowing. Increasing the membrane thickness can reduce bowing and make the window more robust, but resolution will decrease due to the electron scattering with the membrane material itself. Another way is to incorporate micrometre-sized pillars pinning the top and bottom window together [156]. However, this requires the manufacture of top and bottom chips as a single piece, which is more expensive and prone to breaking. The most popular way to reduce bowing is to simply reduce the size of the window.

A good way to mitigate electron scattering by the window is to use alternative membrane materials. For example, graphene-membrane cells achieve better resolution than SiN [47–52]. Recently, $MoS_2$ [157] was also successfully used as the membrane material to see Pt nanoparticle growth. However, as

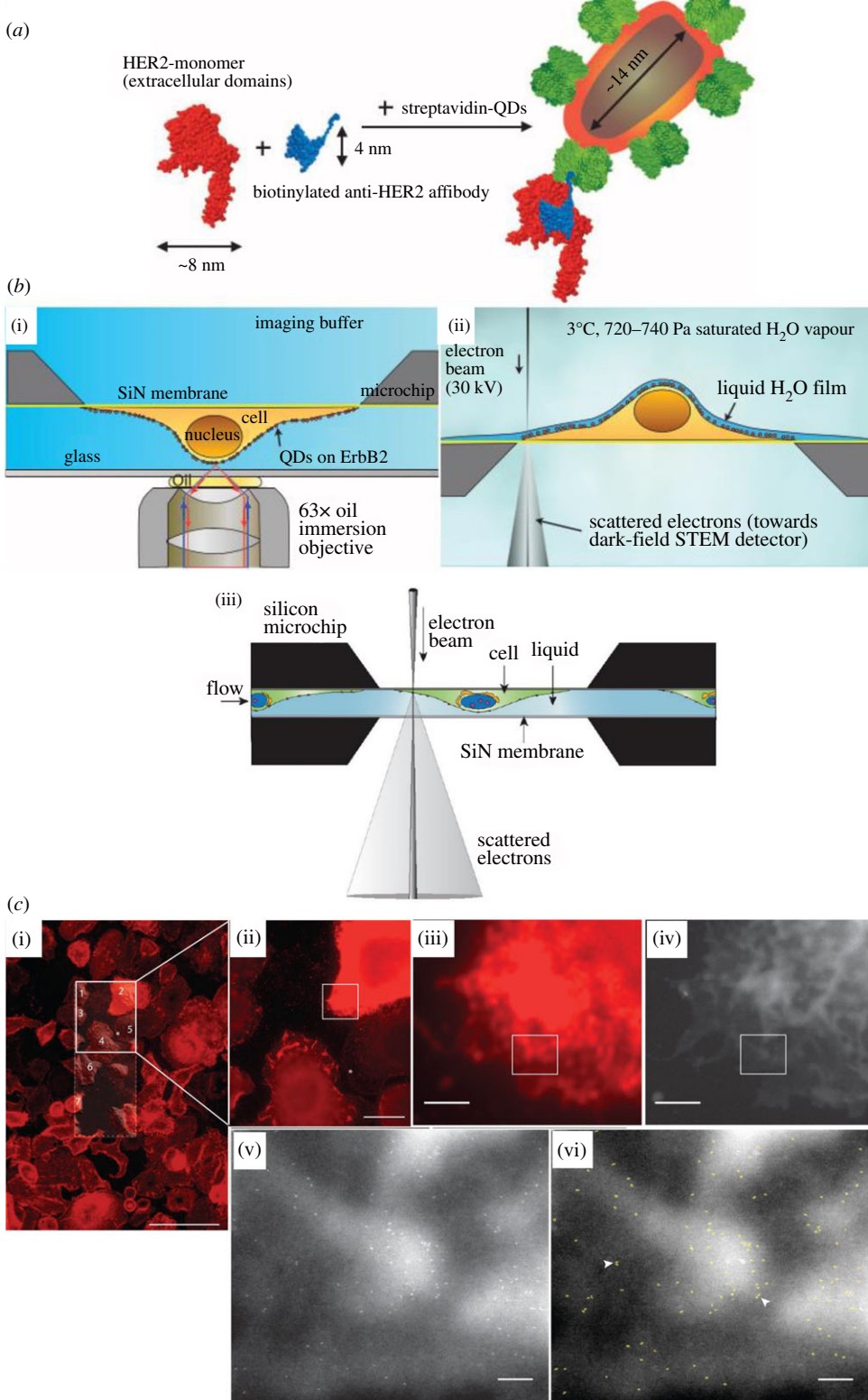

**Figure 6.** (*a*) Schematics of HER2 labelling with affibody-quantum dots [141]. (*b*) Schematics showing correlative microscopy of whole cells in liquid cell. (i) Fluorescence imaging [141]. (ii) For STEM imaging within environmental SEM [141]. (iii) For STEM imaging inside TEM [139]. (*c*) Correlative light and electron microscopic images of HER2-labelled SKBR3 cell [141]. (i) Fluorescence overview image showing several dozen cells. (ii) Higher-resolution fluorescence image from marked solid line rectangle in (i). (iii) Even higher-resolution fluorescence image from selected area in (ii). (iv) STEM image of the same area as (iii). (v) STEM image of the boxed region shown in (iii) and (iv). (vi) Automatically detected labels outlined in light green. Scale bars, 100 µm (i), 20 µm (ii); 2 µm (iii and iv); 200 nm (v and vi).

**Table 2.** Different types of electron beam damage and their effects on liquid cell TEM (dotted lines indicate relatively insignificant contribution).

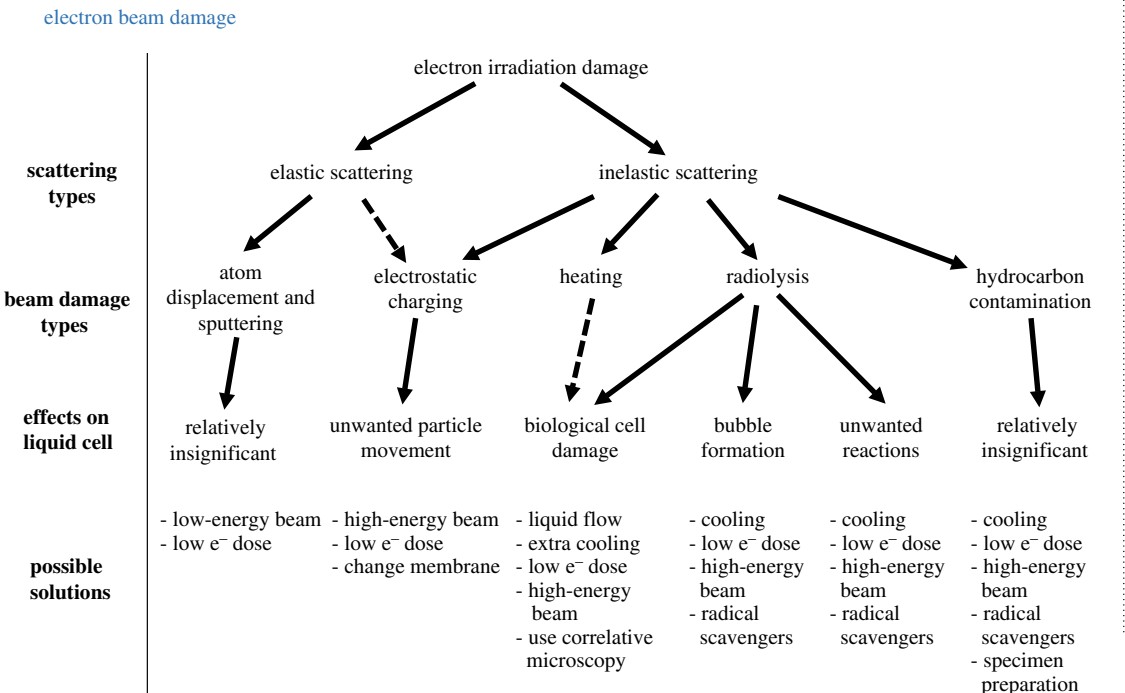

electron beam damage

| | | | | |
|---|---|---|---|---|
| **scattering types** | elastic scattering | | inelastic scattering | |
| **beam damage types** | atom displacement and sputtering / electrostatic charging | heating | radiolysis | hydrocarbon contamination |
| **effects on liquid cell** | relatively insignificant / unwanted particle movement | biological cell damage | bubble formation / unwanted reactions | relatively insignificant |
| **possible solutions** | - low-energy beam<br>- low e⁻ dose | - high-energy beam<br>- low e⁻ dose<br>- change membrane | - liquid flow<br>- extra cooling<br>- low e⁻ dose<br>- high-energy beam<br>- use correlative microscopy | - cooling<br>- low e⁻ dose<br>- high-energy beam<br>- radical scavengers | - cooling<br>- low e⁻ dose<br>- high-energy beam<br>- radical scavengers | - cooling<br>- low e⁻ dose<br>- high-energy beam<br>- radical scavengers<br>- specimen preparation |

mentioned, these membrane materials can only encapsulate a very small amount of liquid, and they are not compatible with microfabrication and therefore functions like electrical biasing and liquid flow. Using correlative techniques like AFM, traction force microscopy (TFM) [158] can also help obtain more information and potentially better resolution.

## 6.2. Electron beam damage

The electron beam can induce many artefacts during the imaging process within a liquid cell [159]. Here we will cover the origin of these artefacts and discuss with respect to the five types of beam damage that commonly occur in electron microscopy, as summarized by Egerton *et al.* [160] and shown in table 2. Both elastic and inelastic electron scattering can lead to beam damage. Elastic scattering results from the beam's interaction with the atomic nucleus, with no energy change of the scattered electron. Inelastic scattering results from the beam's interaction with the electron shells of atoms, with accompanying energy loss that results in secondary electrons, X-ray emission etc.

Atom displacement and sputtering is a type of beam damage due to high-energy, high-angle elastic electron scattering, which displaces atomic nuclei to interstitial positions, or sputters surface atoms away from the specimen. Since liquid cell TEM contains mainly liquid phase specimens, such damage is not as significant as the other types, except potentially when imaging crystalline formation within liquid. A low electron dose rate or lower beam energy can be used to mitigate its effects.

Electrostatic charging can be due to both elastic and inelastic scattering, although inelastic scattering has a dominant effect due to the generation of secondary electrons. Within a liquid cell, charge can build up on the insulating SiN membranes. And such surface charge is worsened by a local enhancement of electron dose formed at the solid–liquid interface which acts as secondary electron source due to scattering [100]. Such surface charge not only interferes with the incoming beam, it can also induce unwanted particle movement within liquid due to repulsion [114]. One of the proposed solutions is to use a very low-energy beam that does not have enough energy to generate secondary electrons. However, such energy limit is usually 50–150 eV [160], which is way too low for any electrons to transmit through the specimen and therefore is not a plausible method. A better solution for TEM is, instead, to use very high energy that has a bigger mean free path for scattering, accompanied by a thin specimen, so most of the beam will transmit through the specimen without generating secondary electrons. Reducing the electron dose rate can also reduce the total charging effect. Another potential

solution is to use a different membrane material. For example, using graphene instead of SiN was reported to reduce such beam damage, allowing higher dose by one magnitude for imaging [161].

Heating is another type of beam damage due to inelastic scattering of electrons where part of their kinetic energy was converted to thermal energy. The relatively large liquid volume within the liquid cell and possible liquid flow system allows good heat exchange to prevent localized heating, which makes heating a relatively insignificant beam damage mechanism in the liquid cell. However, this can still contribute to the degradation of biological cells that are generally very sensitive to localized heating and are normally characterized within an enclosed cell without liquid flow.

A bigger contributor to biological cell degradation is radiolysis due to the electron beam, which destroys the cells [12,138] and induces structural change to biological materials [54]. Keeping a low electron dose is critical to limit the cell damage during TEM imaging. A systematic damage study on the cell by beam dose was performed by Hermannsdörfer *et al.* [162] in 2016. They found a beam dose limit of around $10^2$ electrons $nm^{-2}$ should not be exceeded during imaging [59,162]. It is almost impossible to have good contrast or resolution of organic biological features at such low dose [137]. Therefore, the best way is to use nanoparticle labelling protein [8,12] and use correlative light and electron microscopy [139,141], which should give enough contrast at very low dose.

Radiolysis is the most critical beam damage mechanism in most liquid cell TEM experiments. Radiolysis can lead to bubble formation and expansion, and other unwanted reactions. It has been shown that intensive illumination can lead to bubble formation [163] due to hydrolysis of water and lead to the growth of existing bubbles [164]. Systematic studies of beam-induced gas formation within liquid cell were performed [165,166]. It is well known that the electron beam can induce reactions, which can be beneficial as a way to control the reactions we want to observe [3,6]. However, unwanted reduction of the ions within the solution can lead to crystallization of unwanted nanoparticles [167] which is a common artefact. Other artefacts include particle shape change [108], gelation of ionic liquid [34] etc. These arise from the radical species and aqueous electrons formed within the liquid due to beam irradiation. As mentioned before, these radicals and aqueous electrons are strong reducing species that will react with aqueous electronegative ions and induce or accelerate the reaction [86]. Apart from using a lower dose, radiolysis can also be reduced by reducing temperature and adding scavenger molecules to reduce the radicals.

Hydrocarbon contamination due to inelastic scattering is not well understood within a liquid cell, since it is hard to distinguish its effects from those due to radiolysis, especially on the inside surface exposed to liquid. However, better cell preparation including plasma cleaning, heating and electron beam shower can help get rid of the hydrocarbon contamination on the SiN outside surface exposed to vacuum.

Overall, electron beam damage from inelastic scattering is responsible for most of the artefacts observed during TEM imaging of liquid phase samples. The best way to mitigate their effects is to use a high beam energy (to increase the mean free path and reduce total scattering events, and therefore the generation of secondary electrons) and a low electron dose rate (to reduce total scattering events). It is also important to perform appropriate control experiments where possible, so that beam-induced effects can be distinguished from the experiment of interest.

Thanks to the recent development in (S)TEM, better image contrast can be achieved at a much lower dose. For example, sub-sampling approaches [168,169] allow for high-resolution imaging with very low dose. Acquisition optimization techniques like feature adaptive sampling [170] can also reduce beam damage by reducing unnecessary beam scans to have a lower total beam dose. Further development of low-dose optimized electron microscopy techniques will be essential to realize the objective of *in situ* liquid TEM studies without beam-induced artefacts, and thus an accurate representation of the actual processes that occur in biological cells, batteries and nanostructure synthesis.

## 6.3. Representativeness

Due to the limited solution volume and confined space, in addition to the beam damage mechanisms discussed above, liquid cell TEM results' representativeness is always in doubt. For example, studies of battery electrode materials using liquid cell are currently limited. The electrode materials were either assembled onto the microchips by picolitre drop casting [123] or by flowing a suspension [133]. Not only do these methods have a low success rate, the amount of material deposited onto the chip is too small, which makes such systems far from being representative of their real battery counterparts. To improve its representativeness, a larger amount of material with better attachment to the electrode should be achieved by either nanoscale thin-film patterning [153,154,171] or by focused ion beam (FIB) and nanoscale welding [172]. Membrane materials can also cause problems, making liquid cell TEM less representative.

Donev *et al.* [173] showed that for electron-beam-induced deposition, the membrane material can have a great influence on the morphology and size of the nanoparticles formed. For the Cu deposition and stripping experiment [1], the cyclic voltammogram in liquid cell is quite different from that performed in bulk solution. This is due to the hindered diffusion of liquid and particle within the liquid cell. Such hindered diffusion can also contribute to the unexpected size and number distribution of the deposited material [99]. Nanoparticle diffusion in liquid cell was reported to be up to seven to nine orders slower than expected for a bulk solution [155]. This slowing is due to increase in liquid viscosity (by five orders of magnitude) [110] caused by the formation of an ordered liquid layer due to the surface charge [174] of the silicon nitride windows. A solution for this problem is to either find a way to cancel the charge on the SiN membrane or to use a different membrane material that does not interact or charge as easily.

## 6.4. Other problems

Apart from the three main challenges mentioned above, there are other common issues like trapped bubbles [4,56] that can be solved with better cell design, and sample movement during imaging which can lead to compromised resolution and requires additional post-experiment processing to track individual particles [84]. In this case, immobilization by functionalizing SiN [175,176] can be used to fix material to improve the image stability and resolution. A final problem that is worth noting is the technically demanding nature of performing *in situ* liquid cell TEM. However, the competitive commercial environment for liquid cell TEM holders has led to increasingly user-friendly systems with each generation.

So far, compared with imaging solid specimens using TEM, imaging liquid is still quite limited. There is still great potential to improve the technique. For example, most studies only perform bright/dark field STEM. More information can be potentially extracted if the liquid cell experiments can be further optimized for functions like diffraction, EELS and tomography. The cell design could also be improved so that the final electrode products from the *in situ* experiment could be non-destructively removed from the sample chip, such that they could be analysed using other complementary characterization techniques *ex situ*. Finally, developing TEM holders that allow the sample-holding tip to be detached from the holder would allow complementary experiments to TEM to be performed more easily, as the tip could be loaded in the appropriate instrument.

Data accessibility. This article has no additional data.

Authors' contributions. S.P. and C.G. both wrote the initial review drafts. A.W.R. set the scope, advised on references and finalized the manuscript.

Competing interests. We declare we have no competing interests.

Funding. A.W.R. is funded by a Royal Society University Research Fellowship.

Acknowledgements. A.W.R. would like to thank the support of the Royal Society.

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
