## [Reviewer comments · Royal Society Open Science]

Review History

RSOS-191204.R0 (Original submission)

Review form: Reviewer 1

Is the manuscript scientifically sound in its present form?

Yes

Are the interpretations and conclusions justified by the results?

Yes

Is the language acceptable?

Yes

Do you have any ethical concerns with this paper?

No

Have you any concerns about statistical analyses in this paper?

No

Recommendation?

Accept with minor revision (please list in comments)

Comments to the Author(s)

In this manuscript the authors review how in-situ TEM experiments of liquids can be performed, with a particular focus on microchip-encapsulated liquid-cell TEM. This manuscript covers the basics of the technique, its strengths and weaknesses with respect to related in-situ TEM methods for characterizing liquid systems, and its application in biological, chemical, and materials science fields. This manuscript can guide significance for the development and application of liquid-cell TEM and thus is very important for the related fields. I would like to recommend its publication in Royal Society Open Science after the following minor revisions are done.

1. The description of the 'miniature battery' system is not detailed (page 5, line 50-51, and page 6, line 17-21). Could the authors please give more discussion? For example, how to suspend the ionic liquid on the vacuum and make sure it is thin enough?
2. In page 6, line 31-49, about the open environmental cell or environmental TEM, the author mentioned, "regardless of their vapour pressure... however, in 'open environmental cell'... most liquids with high vapour pressure will still evaporate". This method is very important for liquid-phase TEM experiments, but generally it can be only applied with high-vapour-pressure liquids. Could the authors please give more discussion about "most liquids with high vapour pressure will still evaporate"?
3. In page 26, line 14-15, the author mentioned "Li is a very light and reactive element". Could the authors please give more explanations about Li for its importance of avoiding the dendrite formation? I know this, but the authors had better explain this more detailedly because it is important for general readers.
4. I think it is better for the authors to give more prospects for the development and application of liquid-cell TEM.

Review form: Reviewer 2

Is the manuscript scientifically sound in its present form?

Yes

Are the interpretations and conclusions justified by the results?

Yes

Is the language acceptable?

Yes

Do you have any ethical concerns with this paper?

No

Have you any concerns about statistical analyses in this paper?

No

Recommendation?

Accept as is

Comments to the Author(s)

This manuscript by Robertson et al. reviews the use of liquid cells to characterise materials using transmission electron microscopy. This technique is useful and has huge potential to further develop for many applications. The review is very-well written including the basics of the strengths and the weaknesses of the technique. As such, I recommend publishing this manuscript as is.

Decision letter (RSOS-191204.R0)

04-Oct-2019

Dear Dr Robertson:

Title: Liquid cell TEM and its Applications
Manuscript ID: RSOS-191204

Thank you for submitting the above manuscript to Royal Society Open Science. On behalf of the Editors and the Royal Society of Chemistry, I am pleased to inform you that your manuscript will be accepted for publication in Royal Society Open Science subject to minor revision in accordance with the referee suggestions. Please find the reviewers' comments at the end of this email.

The reviewers and handling editors have recommended publication, but also suggest some minor revisions to your manuscript. Therefore, I invite you to respond to the comments and revise your manuscript. I apologise this has taken longer than usual.

Please also include the following statements alongside the other end statements. As we cannot publish your manuscript without these end statements included, if you feel that a given heading is not relevant to your paper, please nevertheless include the heading and explicitly state that it is not relevant to your work. We have included a screenshot example of the end statements for reference.

- Acknowledgements

- Funding statement

Please include a funding section after your main text which lists the source of funding for each author.

Because the schedule for publication is very tight, it is a condition of publication that you submit the revised version of your manuscript before 13-Oct-2019. Please note that the revision deadline will expire at 00.00am on this date. If you do not think you will be able to meet this date please let me know immediately.

Best wishes,
Dr Laura Smith
Publishing Editor, Journals

On behalf of the Subject Editor Professor Anthony Stace and the Associate Editor Professor Tobias Hertel.

RSC Associate Editor:
Comments to the Author:

(There are no comments.)

RSC Subject Editor:

Comments to the Author:

(There are no comments.)

Reviewer comments to Author:

Reviewer: 1

Comments to the Author(s)

In this manuscript the authors review how in-situ TEM experiments of liquids can be performed, with a particular focus on microchip-encapsulated liquid-cell TEM. This manuscript covers the basics of the technique, its strengths and weaknesses with respect to related in-situ TEM methods for characterizing liquid systems, and its application in biological, chemical, and materials science fields. This manuscript can guide significance for the development and application of liquid-cell TEM and thus is very important for the related fields. I would like to recommend its publication in Royal Society Open Science after the following minor revisions are done.

1. The description of the 'miniature battery' system is not detailed (page 5, line 50-51, and page 6, line 17-21). Could the authors please give more discussion? For example, how to suspend the ionic liquid on the vacuum and make sure it is thin enough?
2. In page 6, line 31-49, about the open environmental cell or environmental TEM, the author mentioned, "regardless of their vapour pressure... however, in 'open environmental cell'... most liquids with high vapour pressure will still evaporate". This method is very important for liquid-phase TEM experiments, but generally it can be only applied with high-vapour-pressure liquids. Could the authors please give more discussion about "most liquids with high vapour pressure will still evaporate"?
3. In page 26, line 14-15, the author mentioned "Li is a very light and reactive element". Could the authors please give more explanations about Li for its importance of avoiding the dendrite formation? I know this, but the authors had better explain this more detailedly because it is important for general readers.
4. I think it is better for the authors to give more prospects for the development and application of liquid-cell TEM.

Reviewer: 2

Comments to the Author(s)

This manuscript by Robertson et al. reviews the use of liquid cells to characterise materials using transmission electron microscopy. This technique is useful and has huge potential to further develop for many applications. The review is very-well written including the basics of the strengths and the weaknesses of the technique. As such, I recommend publishing this manuscript as is.

Author's Response to Decision Letter for (RSOS-191204.R0)

See Appendix A.

Decision letter (RSOS-191204.R1)

19-Nov-2019

Dear Dr Robertson:

Title: Liquid cell TEM and its Applications
Manuscript ID: RSOS-191204.R1

It is a pleasure to accept your manuscript in its current form for publication in Royal Society Open Science. The chemistry content of Royal Society Open Science is published in collaboration with the Royal Society of Chemistry.

The comments of the reviewer(s) who reviewed your manuscript are included at the end of this email. I apologise that this has taken longer than usual.

On behalf of the Subject Editor Professor Anthony Stace and the Associate Editor Professor Tobias Hertel.

RSC Associate Editor
Comments to the Author:
(There are no comments.)

Reviewer(s)' Comments to Author:

Appendix A

To the Editors,

Many thanks for the invitation to respond to the minor revisions. We believe that we have addressed all the reviewer comments, which we have outlined point-by-point below. Along with this resubmission, we have included the corrected manuscript with changed/added sections highlighted in yellow.

If you have any further queries, please do not hesitate to let me know.

Alex Robertson

1. *The description of the 'miniature battery' system is not detailed (page 5, line 50-51, and page 6, line 17-21). Could the authors please give more discussion? For example, how to suspend the ionic liquid on the vacuum and make sure it is thin enough?*

Thank you for pointing that out. More explanation has been added on page 6, it is mainly the anode material being imaged, not the ionic liquid and the anode material is in the form of nanowire, so it is thin enough for TEM.

2. *In page 6, line 31-49, about the open environmental cell or environmental TEM, the author mentioned, "regardless of their vapour pressure... however, in 'open environmental cell'... most liquids with high vapour pressure will still evaporate". This method is very important for liquid-phase TEM experiments, but generally it can be only applied with high-vapour-pressure liquids. Could the authors please give more discussion about "most liquids with high vapour pressure will still evaporate"?*

Thank you for pointing that out. More explanation and examples have been given out in page 6 and 7.

3. *In page 26, line 14-15, the author mentioned "Li is a very light and reactive element". Could the authors please give more explanations about Li for its importance of avoiding the dendrite formation? I know this, but the authors had better explain this more detailedly because it is important for general readers.*

Thank you for pointing that out. More explanation have been given.

4. *I think it is better for the authors to give more prospects for the development and application of liquid-cell TEM.*

We have added another paragraph at the end of the paper

We have also included an "Acknowledgements" and "Funding" statement at the end of the manuscript.